# Evaluation of Autochthonous Coagulase—Negative Staphylococci as Starter Cultures for the Production of Pastırma

**DOI:** 10.3390/foods12152856

**Published:** 2023-07-27

**Authors:** Kübra Fettahoğlu, Mükerrem Kaya, Güzin Kaban

**Affiliations:** 1Doğubayazıt Ahmed-i Hani Vocational School, Ağrı İbrahim Çeçen University, Ağrı 04100, Türkiye; 2Department of Food Engineering, Faculty of Agriculture, Atatürk University, Erzurum 25240, Türkiye; mkaya@atauni.edu.tr (M.K.); gkaban@atauni.edu.tr (G.K.); 3MK Consulting, Ata Teknokent, Erzurum 25240, Türkiye

**Keywords:** pastırma, dry-cured meat product, *Staphylococcus xylosus*, *Staphylococcus equorum*, *Staphylococcus vitulinus*, starter culture

## Abstract

The aim of the study was to investigate the effects of *Staphylococcus xylosus* 39, *S. equorum* 53, or *S. vitulinus* 75, previously isolated from pastırma, on the quality characteristics of pastırma, a Turkish dry-cured meat product, and to evaluate their potential use as starter cultures. The pastırma production was carried out with a traditional method. The control pastırma groups were manufactured without adding any starter culture. At the end of production, the groups were subjected to microbiological and physico-chemical analyses. The pH was above 5.5, and the a_w_ value was below 0.90 in all groups. The strains used exhibited good adaptation to the pastırma. The *S. equorum* 53 decreased the thiobarbituric acid reactive substances (TBARS) value in pastırma, while the *S. xylosus* 39 increased the redness (a*) color value. The autochthonous strains caused a decrease in the palmitic acid (C16:0). However, they had no significant effect on the stearic acid (C18:0) and the oleic acid (C18:1n-9c). A total of 41 volatile compounds were identified in the groups. *S. vitulinus* 75 increased both benzaldehyde and 2-methyl-3-phenylpropanal levels. In addition, the principal component analysis (PCA) of volatile compounds provided a good separation, and PC1 separated *S. xylosus* 39 from other groups.

## 1. Introduction

Pastırma, the most representative Turkish dry-cured meat product, is made by curing, drying, and applying a paste called çemen to whole muscles obtained from certain parts of beef or water buffalo carcasses [1]. The traditional production is carried out under natural conditions depending on climate and weather conditions in late September and October to November [2]. Gram positive, catalase-positive cocci have an important share in the microbiota of pastırma [1,2,3,4]. Coagulase-negative staphylococci (CNS) constitute an important part of these microorganisms [4,5]. It is stated that CNS are technologically important microorganisms in whole-muscle meat products such as ham and pastırma [6,7]. CNS are involved in the formation and stability of color via nitrate reductase activity, which leads to the nitrosomyoglobin formation in cured meat products [8,9,10,11,12]. Nitrite formed by reducing nitrate can also limit lipid oxidation [10,12]. Moreover, these microorganisms play an important role in preventing lipid oxidation by breaking down H_2_O_2_ with catalase and superoxide dismutase activities [10]. On the other hand, CNS play a role in flavor development by forming low molecular weight compounds with their proteolytic and lipolytic activities [7,9,10].

It is very important to choose a starter culture suitable to obtain a quality product. Starter cultures that are not suitable for the product process may adversely affect the sensory properties of the product [13,14,15]. Meat-starter cultures are divided into two categories. The first-generation starter cultures are generally from vegetable fermentation such as *Lactiplantibacillus plantarum* and pediococci. The second-generation cultures such as *Latilactobacillus sakei* and CNS are from meat origins. Therefore, they adapt very well to the meat fermentation conditions [16]. Traditional products can be made without the use of starter cultures. However, industrial production requires a standardization of product properties and accelerated processing [8,9,10,11,12,13,14,15,16,17].

The application of a starter culture is very difficult for whole-muscle meat products such as dry-cured ham, dry-cured loin, and pastırma. It is of great importance that microorganisms reach the inner part of the product without damaging the meat texture [18]. There are studies investigating the possibilities of using commercial starter culture preparations with *Staphylococcus xylosus* and *S. carnosus* for pastırma production [19,20,21,22]. However, there is no study on the impact of autochthonous CNS strains on the quality properties of pastırma as starter cultures. Previous studies have described that *S. xylosus, S. saprophyticus*, *S. equorum* [4,5,6], and *S. vitulinus* [4] are the dominant species in the microbiota of pastırma. Other identified staphylococci are *S. simulans, S. warneri, S. haemolyticus, S. hominis,* and *S. gallinarum* [6].

The natural microbiota in traditional meat products are a good source for the development of starter cultures due to their adaptation to the meat matrix and their ability to compete with the other product microbiota [23,24]. CNS, which are acid-sensitive, can show a good growth due to the pH value of pastırma (>5.5) [4]. Therefore, CNS can be considered as one of the main candidates for possible use as starter cultures in pastırma. The aim of the study was to investigate the effects of *S. xylosus* 39, *S. equorum* 53, or *S. vitulinus* 75, previously isolated from pastırma [4], on the quality characteristics of pastırma, a Turkish dry-cured meat product, and to evaluate their potential use as starter cultures.

## 2. Materials and Methods

### 2.1. Materials

*M. longissimus thoracis et lumborum* was used for the pastırma production. The muscles obtained from two carcasses (36 months old, male cattle, Brown Swiss crossbred) (24 h post mortem) were taken from the Meat and Dairy Board Erzurum, Türkiye.

Fenugreek (*Trigonella foenum graecum* L.) seed flour, fresh garlic, red sweet pepper, and red hot pepper were obtained from local markets in Erzurum, Türkiye.

*S. xylosus* 39, *S. equorum* 53, or *S. vitulinus* 75, previously isolated from pastırma [4] and examined for their technological properties [25] and antibiotic susceptibility [26], were used as starter cultures for the pastırma production.

Each strain to be used in the production of pastırma (*S. xylosus* 39, *S. equorum* 53, or *S. vitulinus* 75) was transferred to 100 mL nutrient broth (Oxoid, Basingstoke, Hampshire, UK) in a 300 mL erlenmeyer flask as 1 mL and then incubated at 37 °C for 24 h in the shaking at 180 rpm (TH-30, Edmund Bühler, Bodelshausen, Germany). After the incubation, the concentrated cells obtained with centrifugation (2500× *g*, 4 °C, and 10 min) (Thermo Fisher Scientific, MR23G, Waltham, MA, USA) were washed 2–3 times with 0.85% (*w*/*v*) sodium chloride (NaCl) solution. The concentrated cell was resuspended with 1 mL 0.85% NaCl solution, and the cell number per mL was determined on Nutrient Agar (Oxoid) [27]. A cell suspension with approximately 1 × 10^10^ CFU/mL was obtained. In view of this result, it was repeated for each strain before each production.

### 2.2. Pastırma Production

Pastırma production was carried out under natural conditions using the traditional method in a local plant (Kadakçıoğlu Meat and Meat Products, Erzurum, Türkiye). After removing fat and connective tissue from the surface, each muscle was divided into two parts, and four meat pieces were obtained from one carcass. The meat pieces obtained from the same carcass were used for a repetition. A total of 8 meat pieces were used for two treatments. In each production, one piece was evaluated as a control group, and strain inoculation was not performed. The other three pieces were applied separately with *S. xylosus* 39, *S. equorum* 53, or *S. vitulinus* 75. In the inoculation, 1 mL of cell suspension (approx. 1 × 10^10^ CFU/mL) was used for each surface of pieces and before use, the cell suspension was resuspended with 20 mL of 0.85% NaCl solution. Then, the suspension prepared was spread on the surface of pieces. After one hour, the pieces were cured with a curing mixture (80 g NaCl, 3 g saccharose, and 0.3 g potassium nitrate per kg meat) at 9–10 °C for 48 h. After surfaces of the pieces were lightly washed with tap water, the pieces were hung for 6 days at 15–25 °C (first drying). Afterward, they were subjected to the first pressing at 10 °C for 20 h, following which they dried again for 5 days at 18–22 °C (second drying). After the drying, second pressing was carried out at 20–22 °C for 1.5–2 h. After this procedure, the third drying was carried out for 5 days at 15–20 °C. Then, dried pieces were put in a bowl of paste (çemen) containing 500 g flour (*Trigonella foenum graecum*) ground from seed, 350 g fresh garlic, 75 g red sweet pepper, 75 g red hot pepper, and 1200 mL water for a day at 10 °C. The paste on the surface of the meat pieces was trimmed to a 3–4 mm thick layer of paste on each piece. Finally, the paste-covered product was dried at 15–20 °C for 10 days.

### 2.3. Determination of the pH, Water Activity (a_w_), and Thiobarbituric Acid-Reactive Substances (TBARS) Values

Ten g of sample was homogenized with 100 mL of distilled water using an Ultra-Turrax (IKA T25, IKA Werke GmbH & Co, Breisgau, Germany). The pH value was measured with a pH meter (Thermo Fisher Scientific, Orion Star, Waltham, MA, USA).

The a_w_ value was determined using a water activity device at 25 °C (Novasina TH-500 a_w_ Sprint, Pfaffikon, Switzerland).

TBARS value was determined using the method given by Lemon [28]. TBARS value was given as mg malondialdehyde (MDA)/kg.

### 2.4. Microbiological Analyses

Twenty-five g samples were weighed into sterile Stomacher bags and added to 225 mL of sterile saline water (0.85% NaCl). The homogenization was performed for 2 min in Stomacher (Lab Stomacher Blander 400-BA, London, UK). Serial dilutions were prepared, and the spread-plate method was used for enumeration of *Micrococcus/Staphylococcus* and *Enterobacteriaceae* counts.

Mannitol Salt Phenol Red (MSA, Oxoid) agar was used for *Micrococcus/Staphylococcus*. Incubation was carried out at 30 °C for 2 days under aerobic conditions. *Enterobacteriaceae* were determined on Violet Red Bile Dextrose (VRBD, Merck, Darmstadt, Germany) agar. The plates were incubated at 30 °C for 2 days under anaerobic conditions.

### 2.5. Determination of Color Values

After the pastırma samples were sliced into slices 1.5 mm in thickness, L*, a*, and b* values were measured with a Chroma Meter (CR-400 Konika Minolta, Osaka, Japan) with an aperture size of 8 mm, a *C D65 illuminant, and standard observed of 2°. All measurements were made in triplicate.

### 2.6. Determination of Fatty Acid Composition

One g of pastırma sample was transferred to the centrifuge tube, and 20 mL of the solvent (2:1 chloroform:methanol + 0.25 g/L butylated hydroxytoluene) was added to it. After it was subjected to homogenization with an Ultra-Turrax homogenizer for 1 min, it was filtered with a nuche Erlenmeyer, and 4 mL MgCl_2_·6H_2_O was added to it. After the sample in the tube was treated with nitrogen, it was vortexed and kept in the dark for 24 h, and the lower phase was transferred [29]. The methylation process of fatty acids was performed according to Metcalfe and Schmitz [30]. The samples taken into vials were kept at −18 °C until analysis. GC/FID (Gas Chromatography/Flame Ionization Detector, Agilent Technologies 6890 N, Wilmington, DE, USA) was used to determine the fatty acid composition. In the system, CPSIL 88 (Agilent 100 m × 250 μm × 0.20 μm) was used as a column. The oven temperature was gradually increased from 100 °C to 225 °C. Helium was used as a carrier gas with a 1.2 mL/min flow rate. A fatty acid methyl ester mix (Supelco, FAME-mix, 47885, Bellefonte, PA, USA) was used as the standard, and the results were given as percentage.

### 2.7. Determination of Volatile Compounds

Five g of sample were weighed into 40 mL vials. After the vials were kept in a 30 °C thermal block for 1 h, the carboxen/polydimethylsiloxane (CAR/PDMS) fiber (Supelco) was inserted into the vial for the extraction of volatile compounds using the solid-phase microextraction technique. The fiber was kept in the vial at the same temperature for 2 h, and the extraction process was completed. Gas chromatography (GC, Agilent Technologies 6890N)/mass spectrometry (MS, Agilent Technologies 5973) was used to identify volatile compounds. DB-624 (J&W Scientific, 60 m × 0.25 mm × 1.4 μm film) was used as the column in the system. The oven temperature was initially at 40 °C for 5 min; then it was gradually increased from 110 °C to 210 °C and kept at this temperature for 12 min. Helium was used as a carrier gas in the system at a flow rate of 1 mL/min. In the identification of compounds, the library of mass spectrometry (NIST, WILEY, and FLAVOR) and standard mix (n-Paraffins Mix, Supelco 44585-U) were used, and the standard mix was also used to determine the Kovats index. The results were given in AU × 10^6^ [1].

### 2.8. Statistical Analyses

The analysis of variance (ANOVA) was performed using a general linear model with treatment as a main effect and replications (blocks) as a random effect in a randomized complete block design. Experiments were carried out in two repetitions. The individual standard deviations of the means are presented in the tables. Differences between the means were determined during Duncan’s multiple range tests (*p* < 0.05). The statistical analyses were performed using the SPSS version 20 statistical program (SPSS Inc., Chicago, IL, USA). Principal component analysis (PCA) was also performed using the Unscrambler software (CAMO version 10.1, Oslo, Norway) to determine the relation between the autochthonous strains and volatile compounds or fatty acid compositions.

## 3. Results and Discussion

### 3.1. pH and a_w_ Values

The highest pH was found in the pastırma with *S. equorum* 53. No significant difference in pH values was observed between the control group and the group with *S. vitulinus* 75 (*p* > 0.05) (Table 1). Similarly, Aksu and Kaya [19] determined that the commercial preparations containing *S. carnosus* did not cause a significant change in the pH value of pastırma. In another study conducted by Bosse et al. [27], the pH of dry-cured ham containing *S. carnosus* LTH 3838 or *S. carnosus* LTH 7036 strains was higher than the control group, but there was no statistically significant difference between the groups. According to the results obtained from the present study, *S. equorum* 53 and *S. xylosus* 39, among the autochthonous staphylococci strains, gave slightly higher average pH values compared to the control, but the pH values measured in all groups are within the acceptable limits [31].

The use of autochthonous strains had a significant effect (*p* < 0.05) on the a_w_ value. The groups containing *S. xylosus* 39 or *S. vitulinus* 75 showed a higher a_w_ value compared to the control group (Table 1). On the contrary, Bosse et al. [27] determined a lower average a_w_ value in dry-cured raw samples containing *S. carnosus* LTH 3838 and *S. carnosus* LTH 7036 strains compared to the control. It was also emphasized that there was no significant difference between *S. carnosus* strains. a_w_ is an important hurdle effect for the microbiological stability of pastırma. However, for sensory reasons, the a_w_ value of pastırma should be between 0.85 and 0.90 [1]. In our study, the a_w_ value was found in this range in all groups.

### 3.2. Thiobarbituric Acid-Reactive Substances (TBARS)

The use of autochthonous strains had a significant effect on the TBARS value of pastırma (*p* < 0.05) (Table 1). The highest mean TBARS value was determined in the group inoculated with *S. vitulinus* 75; however, this value did not differ statistically from the mean values of the control and *S. xylosus* 39 group (*p* > 0.05). The lowest mean TBARS value was determined in the presence of *S. equorum* 53 (*p* < 0.05), and this value did not differ from the average value of the *S. xylosus* 39. Aksu and Kaya [21] determined that a commercial starter culture containing *S. carnosus* showed a low TBARS value during pastırma production. It is thought that the strains used in this study, especially the *S. equorum* 53 strain, showed lower TBARS values compared to the control due to the nitrate reductase and catalase enzyme activities [4]. Talon et al. [32] also stated that staphylococci have an antioxidant property and some strains of *S. xylosus* exhibit superoxide dismutase activity. Moreover, Landeta et al. [12] indicated that the *S.vitulinus* IFJ 4 strain exhibited stronger catalase activity compared to the *S. equorum* IFJ 25 strain. There may be a difference even among different strains of the same species in terms of catalase activity, and *S. xylosus* strains also shows significant differences in terms of catalase activity. The unsaturated fatty acids formed as a result of the activity of lipolytic enzymes are good precursors for autoxidation. In a previous study, the autochthonous strains used in the present study were found to have lipolytic activity [4]. However, it was determined that the strains caused differences in TBARS values. This result showed that the lipolytic activities of the strains are affected differently by the process conditions.

### 3.3. Microbiological Properties

In the study, strains used as starter cultures in pastırma showed good adaptation to production conditions. A higher average count of *Micrococcus*/*Staphylococcus* was determined in the pastırma groups produced using *S. xylosus* 39 and *S. equorum* 53 compared to the control group and the group with *S. vitulinus* 75 (Table 1). As can be seen from Table 1, the *S. vitulinus* 75 strain showed a mean value of 7.26 ± 0.19 log CFU/g. However, this value did not differ statistically from the control group. CNS are the predominant microorganisms in pastırma. These microorganisms affect the sensory properties of pastırma with their catalase, nitrate reductase, proteolytic, and lipolytic activities [3,4,25,26].

The *Enterobacteriaceae* count was found to be below the detectable limit in all groups. It is emphasized that the inability of these family members to survive is due to curing/salting and a decrease in water activity as a result of drying/ripening [1]. Furthermore, in many studies on pastırma, it was reported that the members of the *Enterobacteriaceae* family were below 2 log CFU/g [1,20,33].

### 3.4. Color Values

There was no difference in terms of the mean L* value between the group containing *S. xylosus* 39 and the control group (Table 1). Likewise, the other two strains did not cause a significant change in the L* value compared to the control group (Table 1). In contrast, Bosse et al. [27] reported that the *S. carnosus* LTH 3838 strain caused a decrease in the L* value of dry-cured ham.

CNS are microorganisms effective in color formation with nitrate reductase activities. The group with *S. xylosus* 39 showed the highest average a* value. Similarly, the use of *S. vitulinus* 75 caused an increase in a* value. Bosse et al. [27] also reported that the *S. carnosus* LTH 3838 strain increased the a* value in cured raw ham samples. On the other hand, *S. equorum* 53 did not cause a significant change in the a* value compared to the control (*p* > 0.05). These results can be evaluated as a good indicator that *S. xylosus* 39 and *S. vitulinus* 75 may show strong nitrate reductase activity in the production process of pastırma. It is also known that CNS can eliminate the adverse effects of this compound on color by breaking down hydrogen peroxide with catalase activities. The two most remarkable species in terms of color formation in industrial production are *S. xylosus* and *S. carnosus* [34]. Similarly, Sørensen [35] stated that *S. xylosus*, used as a starter culture in dry-cured meat products, was an important species in terms of color formation and stability. Likewise, Stahnke [36] reported that *S. xylosus*, commonly found in the natural microbiota of fermented sausages, was an important species in terms of color formation by reducing nitrate to nitrite. Prpich et al. [37] also determined that the *S. vitulinus* strain (*S. vitulinus* C2), having nitrate reductase activity, was more effective in the formation of more intense red color compared to the mixed culture (*S. vitulinus* C2 and *L. sakei* 487). In the present study, it is thought that the reason why *S. equorum* 53 strain has the lowest average a* value is probably due to lower nitrate reductase activity compared to others. Indeed, Landeta et al. [12] reported that there was a considerable difference between *S. equorum* strains in terms of nitrate reductase activity. The lowest average b* value in the pastırma samples was found in the group containing *S. equorum* 53, and this average value did not differ statistically from the group without a starter culture (control) (*p* > 0.05). There were statistical differences (*p* < 0.05) between the group with *S. xylosus* 39, in which the highest average b* value was found, and other groups (Table 1). In contrast, Aksu and Kaya [19] stated that the commercial starter culture application did not have an effect on the b* value in the production of pastırma.

### 3.5. Fatty Acid Composition

The use of autochthonous strains showed different effects on the proportions of many fatty acids (Table 2). The highest average values of myristic (C14:0) and myristoleic (C14:1) acids were determined in the group with *S. xylosus* 39, but these average values differed statistically only from the control group (*p* < 0.05) (Table 2). The proportion of palmitic acid (C16:0), which is an important saturated fatty acid in meat and, thus, in pastırma, varied between 26.66 ± 0.94% and 30.92 ± 1.10% in the groups. The control group had the highest mean proportion of palmitic acid (C16:0). There was no statistically significant difference between the groups with autochthonous strains in terms of palmitic (C16:0). The highest proportion of palmitoleic acid (C16:1) was found in the control group (*p* > 0.05) (Table 2). However, no significant difference between the control group and the group with *S. vitulinus* 75 was determined (Table 2). In pastırma groups, the proportion of palmitoleic acid (C16:1) varied between 3.49 ± 0.39% and 2.86 ± 0.54%. Similar results were observed in a study conducted on pastırma by Çakıcı et al. [38].

Palmitic acid (C16:0), stearic acid (C18:0), and oleic acid (C18:1n-9c) were found to be the dominant components of the fatty acids in all pastırma groups (Table 2). The stearic acid (C18:0) in pastırma was not affected by the use of autochthonous strains (Table 2). On the contrary, Aksu and Kaya [21] reported that commercial starter cultures (*S. carnosus, S. carnosus + Lactiplantibacillus pentosus* or *S. xylosus + Latilactobacillus sakei*) caused an increase in this fatty acid. The stearic acid (C18:0) proportions in this study were similar to the results reported in other studies on pastırma [21,38].

Oleic acid (C18:1n-9c), the major fatty acid in pastırma, was not affected by the use of autochthonous strains. The ratio of oleic acid (C18:1n-9c) ranged between 27.87 ± 5.65% and 31.76 ± 2.92% (Table 2). While these results are similar to the results given by Çakıcı et al. [38] (26.15 ± 13.30% and 33.70 ± 7.05%), they were found to be lower from the values provided by Aksu and Kaya [21] (36.19 ± 11.07% and 46.27 ± 3.57%).

Principal component analysis (PCA) was applied to assess the relationships between the strains (*S. xylosus* 39, *S. equorum* 53, and *S. vitulinus* 75) and the fatty acid composition (Figure 1). The first PC was enough to explain 90% of the variation for fatty acid composition. In other words, the first two principal components explained 98% of the total variance (Figure 1). The control and the group with *S. equorum* 53 placed on the positive side of PC1, while the group with *S. xylosus* 39 and *S. vitulinus* 75 placed on the negative side of PC1 (Figure 1). As can be seen in Table 2, there is no difference between the control group and *S. equorum* 53 in terms of myristic (C14:0) and myristoleic (C14:1) acids (Table 2). According to the results of PCA, the use of different strains caused changes in fatty acids. The myristic (C14:0) and myristoleic (C14:1) acids were more related to using *S. xylosus* 39 and *S. vitulinus* 75 in the negative side of PC1. The palmitic acid (C16:0), stearic acid (C18:0), and oleic acid (C18:1n-9c) were located on the positive side of PC1 and showed a positive correlation with the control and *S. equorum* 53 (Figure 1).

### 3.6. Volatile Compounds

In the pastırma groups, 41 volatile compounds belonging to a total of 11 different groups, including 6 sulfur compounds, 2 alcohols, 6 ketones, 8 aliphatic hydrocarbons, 2 esters, 10 aldehydes, 2 aromatic hydrocarbons, 1 nitrogen compound, 2 furans, 1 terpene, and 1 acid, were identified. The use of autochthonous strains did not show a statistically significant difference on sulfur compounds (*p* > 0.05) (Table 3). In the study conducted by Kaban [1] on determining volatile compounds in pastırma, allyl methyl sulfide andmethyl-2-propenyl disulfide compounds were determined, and it was emphasized that garlic in the fenugreek composition was the source of many sulfur compounds. Sulfur compounds may also originate from methionine, cysteine, and cystine [39].

Alcohols that may occur as a result of lipid oxidation, carbohydrate metabolism, and amino acid degradation [33] have a higher threshold value than aldehydes (except 1-octen-3-ol and 1-penten-3-ol) and have little effect on flavor [40]. Ethanol and 1-propen-2-ol were identified in the study, and there was no statistical difference between the groups in terms of these compounds (*p* > 0.05).

Ketones are one of the main groups in dry-cured meat products such as loin and ham [41]. The highest average values of 2,3-butanedione and 3-hydroxy-2-butanone were determined in the group with the *S. equorum* 53 (*p* < 0.05). This group was followed by *S. vitulinus* 75 (Table 3). These compounds were also determined in the dry sausage model system containing *S. equorum* 19 and *S. saprophyticus* 11 [42]. In a study comparing the aroma production of *S. xylosus*, *S. carnosus*, and *S. equorum* in model systems, it was stated that *S. xylosus* and *S. equorum* cause an increase in the levels of methyl-branched ketone as a result of the degradation of leucine, isoleucine, and valine amino acids [43]. Aliphatic hydrocarbons do not have a significant effect on aroma in dry-cured meat products due to high threshold values [1]. In the current study, the use of autochthonous strains did not have a significant effect on aliphatic hydrocarbons (*p* > 0.05) (Table 3). In the study, propyl hexanoate and hexyl butanoate compounds were identified; however, no significant differences were observed between the groups (*p* > 0.05) (Table 3).

Aldehydes constitute an important group of aroma compounds due to their low threshold values. These compounds form an important part of the volatile profile in pastırma [1,33]. In the current study, among aldehydes, acetaldehyde, hexanal, and nonanal were the most abundance compounds (Table 3). Hexanal, a dominant compound in pastırma, is formed from the oxidation of omega-6 fatty acids (linoleic and arachidonic acid), and its high concentrations cause a rancid taste in meat products [2]. *S. vitulinus* 75 showed the highest values for both benzaldehyde and 2-methyl-3-phenylpropanal (Table 3). According to these results, *S. vitulinus* 75 is an important strain for both benzaldehyde and 2-methyl-3-phenylpropanal. Moreover, 3-methyl butanal, which was also determined in the pastırma groups (Table 3), is a highly effective flavoring compound in meat products such as dry sausage and ham [42].

There was no statistical difference in terms of aromatic hydrocarbons between the control group and the groups containing *S. xylosus* 39, *S. equorum* 53, and *S. vitulinus* 75 strains (*p* > 0.05). The sources of aromatic hydrocarbons can vary considerably. In dry-cured meat products, among aromatic hydrocarbons, toluene has an important effect on aroma [1], and it can come from animal feed or may occur as a result of lipid degradation and amino acid catabolism [33].

No difference was determined between the groups in terms of the 2-pentylfuran (*p* > 0.05). This compound has also been detected in various dry-cured meat products [33,44]. As a terpene compound, only limonene was identified in this study. There is no statistical difference between the groups in terms of this compound (*p* > 0.05). Terpenes are related to spices, especially red pepper, and may also originate from herbs used in animal nutrition [39]. There was no statistical difference between the groups in terms of acetic acid (*p* > 0.05), which was the only compound identified as an acid (Table 3).

PCA was applied to evaluate the relationships between the groups (*S. xylosus* 39, *S. equorum* 53 and *S. vitulinus* 75) and volatile compounds (Figure 2). The first two principal components explained 82% (53% for PC1 and 29% for PC2) (Figure 2). PC1 was positively related to the control, *S. vitulinus* 75, and *S. equorum* 53, while *S. xylosus* 39 showed a negative correlation with PC1. On the other hand, the control group separated the groups containing strains in PC2. *S. xylosus* 39, *S. vitulinus* 75, and *S.equorum* 53 groups located on the negative side of PC2, while the control group placed on the positive side of PC2. In addition, volatile compounds were usually collected around *S. vitulinus* 75 in the positive side of PC1 (Figure 2). *S. vitulinus* 75 generally showed a higher abundance in terms of volatile compounds. Statistically higher levels of benzaldehyde and 2-methyl-3-phenyl propanal were determined in this group compared to other groups (Table 3).

## 4. Conclusions

*S. xylosus* 39 and *S. vitulinus* 75 increased the redness (a* value) of pastırma. *S. equorum* 53, on the other hand, decreased the TBARS value, which is a measure of lipid oxidation, compared to the control group. In addition, all autochthonous strains used as starter cultures decreased the level of palmitic acid, which has an important part in the saturated fatty acids of pastırma. According to the PCA result, the fatty acid composition of *S. xylosus* 39 and *S. vitulinus* 75 showed greater similarity. On the other hand, the use of autochthonous strains in pastırma production led to a partial difference in the profile of volatile compounds, and the result of PCA showed that the formation of many volatile compounds was positively correlated with the presence of *S. vitulinus* 75.

## Figures and Tables

**Figure 1 foods-12-02856-f001:**
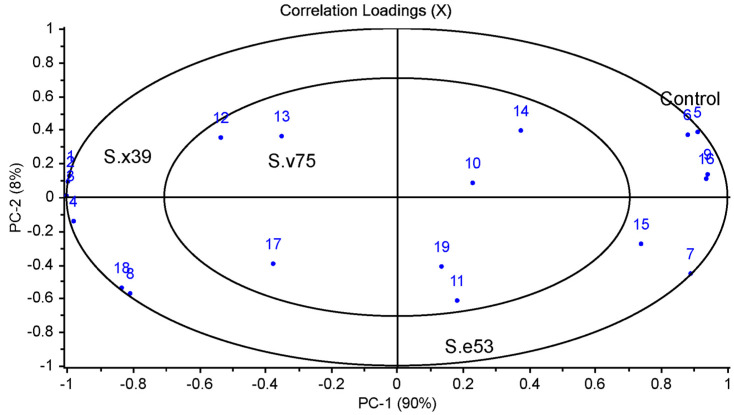
Correlation loadings of the relationships between the groups with autochthonous strains and fatty acid composition (Control: without starter culture, S.x39: *S. xylosus* 39, S.e53: *S. equorum* 53, S.v75: *S. vitulinus* 75, the numbers were given in Table 2).

**Figure 2 foods-12-02856-f002:**
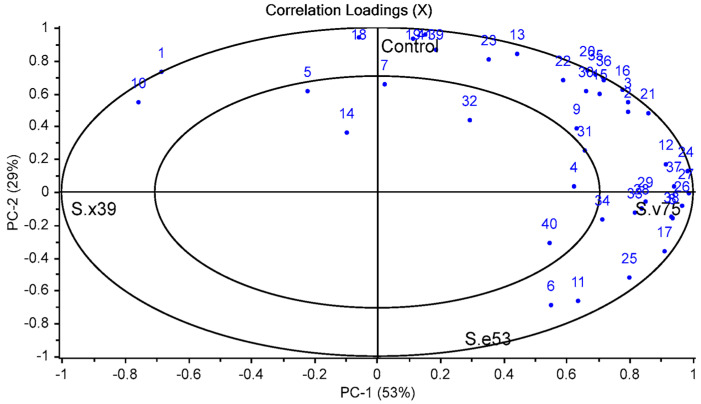
Correlation loadings of the relationships between the groups with autochthonous strains and volatile compounds (Control: without starter culture, S.x39: *S. xylosus* 39, S.e53: *S. equorum* 53, S.v75: *S. vitulinus* 75; compound number was given in Table 3).

**Table 1 foods-12-02856-t001:** The pH, a_w_, TBARS (mg MDA/kg), L*, a*, and b* values and *Micrococcus*/*Staphylococcus* and *Enterobacteriaceae* counts (log CFU/g) of pastırma produced using autochthonous strains (mean ± standard deviation).

Properties	Strains			
	Control	*S. vitulinus* 75	*S. equorum* 53	*S. xylosus* 39
pH	5.63 ± 0.01 c	5.63 ± 0.02 c	5.76 ± 0.01 a	5.66 ± 0.03 b
a_w_	0.866 ± 0.005 b	0.878 ± 0.008 a	0.871 ± 0.001 ab	0.874 ± 0.003 a
TBARS	1.95 ± 0.19 a	2.00 ± 0.06 a	1.68 ± 0.11 b	1.82 ± 0.13 ab
L*	34.25 ± 0.98 ab	35.96 ± 1.31 a	33.06 ± 2.94 b	34.03 ± 1.16 ab
a*	28.70 ± 3.11 c	33.59 ± 1.99 b	27.86 ± 1.24 c	35.98 ± 1.59 a
b*	12.85 ± 1.00 c	16.61 ± 1.37 b	12.48 ± 1.24 c	18.20 ± 0.95 a
*Micrococcus*/*Staphylococcus*	6.94 ± 0.44 b	7.26 ± 0.19 b	7.85 ± 0.08 a	7.81 ± 0.31 a
*Enterobacteriaceae*	<2	<2	<2	<2

a–c: The averages marked with different letters in the same row are statistically different from each other (*p* < 0.05).

**Table 2 foods-12-02856-t002:** The fatty acid composition (%) of pastırma produced using autochthonous strains.

No	Fatty Acid	Strain			
		Control	*S. vitulinus* 75	*S. equorum* 53	*S. xylosus* 39
1	C14:0	2.41 ± 0.04 b	6.00 ± 1.34 a	4.10 ± 1.36 ab	6.80 ± 2.92 a
2	C14:1	1.75 ± 1.00 b	4.77 ± 0.95 a	3.29 ± 0.76 ab	5.54 ± 2.79 a
3	C15:0	0.92 ± 0.24 c	2.79 ± 0.55 ab	2.02 ± 0.53 b	3.17 ± 1.06 a
4	C15:1	0.16 ± 0.07 b	0.56 ± 0.23 a	0.44 ± 0.26 a	0.58 ± 0.05 a
5	C16:0	30.92 ± 1.10 a	26.82 ± 1.10 b	27.04 ± 2.23 b	26.66 ± 0.94 b
6	C16:1	3.49 ± 0.39 a	3.16 ± 0.27 ab	3.03 ± 0.18 b	2.86 ± 0.54 b
7	C18:0	18.92 ± 0.74 a	16.27 ± 1.67 a	18.85 ± 5.70 a	15.76 ± 0.64 a
8	C18:1 n-9 t	0.93 ± 0.21 a	1.01 ± 0.04 a	1.05 ± 0.08 a	1.05 ± 0.12 a
9	C18:1 n-9 c	31.76 ± 2.92 a	29.74 ± 2.26 a	29.71 ± 1.94 a	27.87 ± 5.65 a
10	C18:2 n-6 t	0.15 ± 0.04 ab	0.20 ± 0.06 a	0.14 ± 0.02 ab	0.09 ± 0.03 b
11	C18:2 n-6 c	6.57 ± 0.25 a	6.12 ± 0.61 a	6.88 ± 0.73 a	6.69 ± 0.59 a
12	C20:0	0.12 ± 0.02 a	0.08 ± 0.04 a	0.08 ± 0.02 a	0.33 ± 0.27 a
13	C18:3 n6	0.05 ± 0.03 a	0.24 ± 0.28 a	0.03 ± 0.02 a	0.05 ± 0.03 a
14	C20:1	0.17 ± 0.02 a	0.18 ± 0.07 a	0.15 ± 0.05 a	0.14 ± 0.02 a
15	C18:2 c9 t11	0.56 ± 0.09 a	0.50 ± 0.10 a	0.55 ± 0.06 a	0.53 ± 0.05 a
16	C18:2 t9 t11	0.06 ± 0.02 a	0.05 ± 0.00 a	0.05 ± 0.01 a	0.04 ± 0.01 a
17	C22:0	0.26 ± 0.04 a	0.25 ± 0.07 a	0.28 ± 0.09 a	0.29 ± 0.06 a
18	C22:1 n9	0.60 ± 0.66 a	1.12 ± 0.32 a	1.32 ± 0.33 a	1.34 ± 0.21 a
19	C22:6 n3	0.21 ± 0.02 a	0.18 ± 0.02 b	0.22 ± 0.04 a	0.22 ± 0.02 a

a–c: The averages shown with different letters in the same row are statistically different from each other (*p* < 0.05).

**Table 3 foods-12-02856-t003:** The effects of autochthonous strains on volatile compounds in pastırma.

				Strains			
No *	Compounds	R	KI	Control	*S. vitulinus* 75	*S. equorum* 53	*S. xylosus* 39
	**Sulfur compounds**						
5	Allyl mercaptan	b	574	7.70 ± 2.81 a	8.00 ± 5.37 a	5.40 ± 2.44 a	7.97 ± 2.59 a
8	Allyl methyl sulfide	b	730	5.89 ± 2.77 a	8.59 ± 3.92 a	6.73 ± 2.61 a	4.76 ± 2.30 a
10	Isothiazole	c	772	0.45 ± 0.39 a	0.30 ± 0.35 a	0.17 ± 0.26 a	0.54 ± 0.71 a
17	3,3’-thiobis-1-propene	b	888	16.15 ± 6.97 a	23.52 ± 10.08 a	20.63 ± 9.10 a	14.10 ± 2.98 a
22	Methyl 2-propenyl disulfide	b	958	4.36 ± 2.62 a	4.55 ± 2.34 a	3.42 ± 2.29 a	3.54 ± 1.72 a
33	Di-2-propenyl disulfide	b	1157	29.23 ± 10.80 a	29.15 ± 17.14 a	31.85 ± 8.46 a	23.65 ± 5.39 a
	**Alcohols**						
2	Ethanol	a	527	0.82 ± 0.33 a	0.95 ± 0.33 a	0.61 ± 0.37 a	0.53 ± 0.35 a
3	1-propen-2-ol	c	539	3.83 ± 2.32 a	4.26 ± 2.40 a	2.68 ± 1.27 a	2.19 ± 1.55 a
	**Ketones**						
6	2,3-butanedione	c	657	4.66 ± 3.97 b	7.44 ± 2.64 b	12.54 ± 4.20 a	3.83 ± 1.78 b
11	3-hydroxy-2-butanone	a	779	0.94 ± 0.70 c	2.11 ± 0.63 b	3.50 ± 0.99 a	0.56 ± 0.59 c
20	2-heptanone	c	948	1.66 ± 1.43 a	1.51 ± 1.66 a	0.82 ± 0.61 a	0.59 ± 0.58 a
25	2,3-octanedione	b	1025	10.3 ± 10.13 a	13.6 ± 10.86 a	15.51 ± 10.51 a	8.28 ± 5.8 a
27	6-methyl-5-heptene-2-one	c	1031	0.66 ± 1.42 a	1.02 ± 1.29 a	0.73 ± 0.83 a	0.31 ± 0.42 a
35	3,5-octadiene-2-one	c	1165	2.61 ± 2.88 a	2.33 ± 2.58 a	1.26 ± 0.46 a	0.76 ± 0.48 a
	**Aliphatic hydrocarbons**						
4	Hexane	a	600	5.21 ± 2.19 a	12.78 ± 14.37 a	4.36 ± 1.84 a	5.16 ± 1.91 a
15	Octane	a	800	2.62 ± 1.54 a	1.74 ± 2.15 a	1.47 ± 1.65 a	0.07 ± 0.13 a
18	Nonane	a	900	0.60 ± 0.46 a	0.34 ± 0.25 a	0.32 ± 0.15 a	0.35 ± 0.24 a
23	Decane	a	1000	2.53 ± 2.13 a	1.17 ± 1.61 a	1.14 ± 0.84 a	0.55 ± 0.16 a
31	Undecane	a	1100	2.76 ± 2.14 a	2.19 ± 2.34 a	2.67 ± 2.58 a	1.49 ± 1.44 a
36	Dodecane	a	1200	5.73 ± 2.54 a	5.71 ± 2.93 a	4.36 ± 2.63 a	3.93 ± 1.54 a
39	Tridecane	a	1300	5.59 ± 5.24 a	2.78 ± 2.38 a	2.84 ± 1.74 a	2.27 ± 1.50 a
41	Tetradecane	a	1400	2.59 ± 2.39 a	1.48 ± 0.97 a	1.10 ± 0.04 a	1.19 ± 0.70 a
	**Esters**						
32	Propyl hexanoate	b	1151	10.74 ± 12.17 a	2.99 ± 1.99 a	8.06 ± 6.73 a	1.71 ± 2.00 a
37	Hexyl butanoate	b	1221	6.04 ± 4.83 a	6.39 ± 4.36 a	6.44 ± 2.66 a	3.76 ± 3.12 a
	**Aldehydes**						
1	Acetaldehyde	a	< 500	31.92 ± 8.17 a	22.72 ± 2.73 a	22.15 ± 3.22 a	30.46 ± 10.56 a
9	3-methyl-butanal	c	693	3.40 ± 1.54 a	1.98 ± 2.28 a	2.80 ± 3.14 a	0.44 ± 0.28 a
13	2-methyl-2-butenal	c	788	3.65 ± 1.33 a	2.97 ± 1.79 a	2.73 ± 1.57 a	2.43 ± 1.07 a
16	Hexanal	a	849	37.33 ± 11.44 a	37.19 ± 12.17 a	27.91 ± 14.90 a	22.6 ± 7.70 a
21	Heptanal	b	955	6.04 ± 2.59 a	6.83 ± 1.63 a	4.39 ± 3.44 a	3.04 ± 1.95 a
26	Benzaldehyde	a	1026	6.83 ± 4.56 bc	11.10 ± 3.30 a	7.97 ± 1.81 b	4.04 ± 1.95 c
28	Octanal	a	1054	4.40 ± 2.30 a	5.56 ± 3.44 a	4.57 ± 2.65 a	4.16 ± 2.11 a
34	Nonanal	a	1163	9.61 ± 3.20 a	11.73 ± 3.07 a	9.95 ± 1.39 a	9.67 ± 2.12 a
38	2-nonenal	c	1224	1.16 ± 0.91 a	1.33 ± 0.21 a	1.40 ± 0.81 a	0.70 ± 0.55 a
40	2-methyl-3-phenyl propanal	b	1334	1.21 ± 0.92 b	2.79 ± 1.23 a	1.62 ± 0.77 b	1.66 ± 0.48 b
	**Aromatic hydrocarbons**						
14	Toluene	a	791	0.99 ± 0.60 a	0.53 ± 0.43 a	0.88 ± 0.66 a	0.66 ± 0.57 a
30	1-methyl-2-(1-methylethyl)-benzene	c	1062	1.50 ± 1.02 a	1.26 ± 0.73 a	1.23 ± 0.70 a	0.91 ± 0.37 a
	**Nitrogenous compound**						
12	1-methyl-1H-pyrrole	b	786	1.56 ± 0.68 a	1.56 ± 1.33 a	1.56 ± 0.86 a	1.06 ± 0.63 a
	**Furans**						
19	2-butylfuran	c	925	0.30 ± 0.36 a	0.23 ± 0.23 a	0.23 ± 0.27 a	0.22 ± 0.23 a
24	2-pentylfuran	c	1021	1.86 ± 1.35 a	2.51 ± 1.75 a	1.81 ± 1.08 a	0.9 ± 0.55 a
	**Terpene**						
29	D-Limonene	b	1055	0.85 ± 0.73 a	1.25 ± 0.67 a	0.89 ± 0.46 a	0.74 ± 0.60 a
	**Acid**						
7	Acetic acid	a	717	1.05 ± 0.32 a	1.11 ± 0.80 a	0.77 ± 0.42 a	1.02 ± 0.90 a

a, b: Different letters indicate statistical difference (*p* < 0.05) in each line. SD: Standard deviation. Results are expressed in Arbitrary Area Units (×10^6^) as means of three replicates of each sample. *: compounds no; R: reliability of identification; a: mass spectrum and retention time identical with an authentic sample; b: mass spectrum and Kovats index from the literature in accordance; c: tentative identification with mass spectrum. KI: Kovats index calculated for DB-624 capillary column (60 m × 0.25 mm i.d.x 1.4 μm film) installed on a gas chromatograph equipped with a mass selective detector.

## Data Availability

The data presented in this study are available on request from the corresponding author.

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
