# Peer review of "Evaluation of Autochthonous Coagulase—Negative Staphylococci as Starter Cultures for the Production of Pastırma"

_foods, 2023, doi:10.3390/foods12152856_

Round 1
Reviewer 1 Report (Previous Reviewer 3)
foods-2523577
Authors have corrected the manuscript; however, some additional minor revisions need to be considered.
Ln78 and other occasions: Please, strain identifications (39, 53, 75) do not need to be in italics. Please, check entire manuscript for similar adjustments.
Ln 89: please, delete Basingstoke Hampshire, UK. Information was already provided on Ln82. Please, check the entire manuscript for similar adjustments.
Ln117-120: As previously mentioned, a contact pH meter supposed to be used, since if you mix a sample with water, the pH value recorded will be that of the suspension, and not of the sample. However, looks like the authors did the measurement in the wrong way. For this existing contact pH meters to measure pH of the solid samples.
Ln123: Some details regarding the applied approach need to be provided, not only reference.
In my opinion discussion can be upgraded, however, if the other reviewers are happy with current version, I will accept it as is it.
Author Response
-Authors have corrected the manuscript; however, some additional minor revisions need to be considered.
Ln78 and other occasions: Please, strain identifications (39, 53, 75) do not need to be in italics. Please, check entire manuscript for similar adjustments.
-Revised and checked
Ln 89: please, delete Basingstoke Hampshire, UK. Information was already provided on Ln82. Please, check the entire manuscript for similar adjustments.
-Revised and checked
Ln117-120: As previously mentioned, a contact pH meter supposed to be used, since if you mix a sample with water, the pH value recorded will be that of the suspension, and not of the sample. However, looks like the authors did the measurement in the wrong way. For this existing contact pH meters to measure pH of the solid samples.
pH measurement can be also carried out in homogenized samples. some references:
- De Ávila, M. D. R., Cambero, M. I., Ordóñez, J. A., de la Hoz, L., & Herrero, A. M. (2014). Rheological behaviour of commercial cooked meat products evaluated by tensile test and texture profile analysis (TPA). Meat Science, 98(2), 310-315.
-Hampikyan, H., & Ugur, M. (2007). The effect of nisin on L. monocytogenes in Turkish fermented sausages (sucuks). Meat Science, 76(2), 327-332.
-Karakosta, L. K., Vatavali, K. A., Kosma, I. S., Badeka, A. V., & Kontominas, M. G. (2022). Combined effect of chitosan coating and Laurel essential oil (Laurus nobilis) on the microbiological, chemical, and sensory attributes of water Buffalo meat. Foods, 11(11), 1664.
-Garcia-Rey, R. M., Garcıa-Garrido, J. A., Quiles-Zafra, R., Tapiador, J., & De Castro, M. L. (2004). Relationship between pH before salting and dry-cured ham quality. Meat Science, 67(4), 625-632.
Ln123: Some details regarding the applied approach need to be provided, not only reference.
- The method was shortened due to suggestion of other reviewer.
In my opinion discussion can be upgraded, however, if the other reviewers are happy with current version, I will accept it as is it.
-Thank you.

Reviewer 2 Report (Previous Reviewer 4)
The authors have adequately addressed all the general and specific comments I made in the original manuscript.
Author Response
The authors have adequately addressed all the general and specific comments I made in the original manuscript.
-Thank you.

This manuscript is a resubmission of an earlier submission. The following is a list of the peer review reports and author responses from that submission.
Round 1
Reviewer 1 Report
--------------------------------------------------------------
Reviewer 2 Report
The authors aimed to investigate the effect of three Staphylococcus species on the sensorial and microbiological properties of pastirma, a Turkish dry cured meat product. To this end, they carried out classical physico-chemical and microbiological analyses.
The methods are described, however several remarks can be done in this part. Concerning the strains, the properties of the strains did not need to be described in details, in particular, the results already published such as for example resistance to antibiotics. Futhermore lines 91 to 100 should be included in the paragraph strains. Also in this sentence « In the present study, Staphylococcus xylosus 39, S. equorum 53, and S. vitulinus 75 » replace and by or. TBARS value was determined using the method given by Lemon [28]. As this method is published, the description of the method can be simplified. The microbiological analyses concerned only numeration of two populations (Cocci gram positive, catalase positive and enterobacteriaceae). No numeration after inoculation and after the thorough washing step during the preparation of the pastirma was mentionned. Above all, no monitoring by molecular methods of the Staphylococcus species inoculated has been carried out.
Concerning the paragraph results, it should be written results an discussion. First the numerations of cocci in the control and the three inoculated final products are not so different, meaning that indigenous cocci are present in meat. The cocci population should have been evaluated after the washing step to check that the inoculum was still present. Furthermore, no data are shown to prove the identity of the species in each of the four samples, uninoculated and the three inoculated. We do not know if the cocci species of the four samples are different or not. In fact the pH, aw, TBARS and colour of the four products are very close (Table 1) also the composition in fatty acids and volatile compounds are quite similar (tables 2 and 3). The PCA analysis in Figure 1 on fatty acids should be described in addition to Table 2 and Figure 2 should be discussed with Table 3.
In conclusion, the authors aimed to investigate the effect of three Staphylococcus species on pastirma quality, but the inoculated species were not monitored using molecular tools during the process. This calls into question the validity of the results and the purpose of the study.
Reviewer 3 Report
The present work is interesting. In the pdf provided by the journal I have mentioned several points that merit attention from the authors , will need to be corrected, adjusted, better explained. Some of the principal problems are stated below. However, in revision of the manuscript, I am expecting that authors will cover all problems mentioned in the pdf version of the manuscript. Some additional comments: Authors have applied bacterial cultures, isolated and identified in their previous work. However, description of the strains will be more appropriate to be presented in the introduction and not as part of the material and methods. Since this was previously performed work, will be better to be part of the known information and placed into the introductions section. In the material and methods, authors can only refer to the culturing conditions (media and temperatures), an information relevant to current study. Preparation of the meat product is presented with sufficient details and it is easy to follow the experimental procedure and applied steps. Maybe will be relevant, if authors can provide a reference, where this procedure was standardized according to the food production regulations in Turkey. For material, ingredients, and equipment, please, provide suppliers according to the recommendations from the journal. Normally this needs to include company and address. However, in second occasions for the materials/equipment’s from same company, only name of the supplier will be sufficient. Ln124: If you do a suspension of the peat in the water, pH will be representing the mixture. For the food samples exist contact pH meter that determine the direct pH of the samples and not that of the suspension. Ln130: Please, explain the abbreviation TBARS. Ln144 and etc. Please, consult with colleague and consider to change from "physiological water" to "saline" Please, change to Results and Discussion. If the journal agrees, the both section can be combined. However, the discussion section of the paper will need to be extended a bit more. Several of the obtained results are not really discussed, neither compared with similar studies. Strongly recommend that authors can enrich the discussion section a bit more.

Reviewer 4 Report
GENERAL COMMENTS
The study investigated the quality properties of pastırma inoculated with three strains of Staphylococcus to evaluate their potential for use as starter cultures. The authors produced pastırma using the traditional method. They analyzed the pH, aw, microbiological analyzes, TBARS, fatty acid profile, and volatile compound composition at the end of production. They report that the strains used exhibited good adaptation to the pastırma but caused variable effects on the parameters analyzed at the end of the process. In addition, the authors highlight that the strains used have a limited impact on the aroma profile of pastırma but claim that the strains used can improve the quality properties of pastırma.
The subject of study is of regional interest. The authors must expand the introduction to highlight the importance of their work concerning similar products or, in general, in meat processing research.
The manuscript is well structured, and the conclusions are adequately supported. English is, in general, good, but there are numerous typos. In addition, there are missing spaces and plenty of parentheses, so writing needs to be more careful. Finally, it is recommended a careful review of the document wording.
SPECIFIC COMMENTS
Lines 2-3. The names of the bacteria should be written completely in the title. It is recommended to change the title to "Evaluation of autochthonous Staphylococcus xylosus, Staphylococcus equorumand, and Staphylococcus vitulinus strains as starter culture in pastırma"
Line 52. At the beginning of this line, it is missing a "space" between the words "the" and "application." There are several similar cases. Please carefully review this aspect of the wording throughout the document.
Line 80. Please remove an extra parenthesis from "(Oxoid))"
Line 106. This line mentions 10E10 CFU/mL, but in Line 99, the cell concentration is 1E10 CFU/mL. Please clarify.
Line 405-407. The first two sentences of this paragraph do not come from the results obtained or their analysis. Therefore, they are not a consequence of this work and should not be included in the conclusions.
Reviewer 5 Report
The authors of manuscript FOODRES-2309773 report on the quality properties of pastırma inoculated with Staphylococcus xylosus 39, S. equorum 53, and S. vitulinus 75 isolated from pastırma, a Turkish dry-cured meat product, and to evaluate their potential for use as starter cultures. This topic is of great significance for improving the quality properties of pastırma. However, there is still some modification to be done.
First of all, the origin of beef should be clearly indicated in “Materials and methods”. Please pay attention to adding.
Then, the language of the manuscript is difficult to understand and needs further revision. For example, in the section “results”, “The use of autochthonous strains had an effect on myristic (C14:0), myristoleic (C14:1), palmitic(C16:0) and palmitoleic (C16:1) fatty acids at the level of P<0.05.” It is difficult to find out whether there are significant differences in fatty acids between autochthonous strains or between autochthonous strains and the control group. Please modify them.
Finally, some proper terms in the manuscript are difficult to understand and can be explained a little. For example, “TBARS values” in the section "Materials and Methods".
Specific comments:
Line 78: Spaces should be added between values and units. Elsewhere in the manuscript, when it comes to numbers and units, pay attention to the space between numbers and units.
Line 199-201: The conclusion is inconsistent with the statement in Table 1. There are significant differences among the three indigenous yeasts in the table, but there is no statistical difference in the expression of the conclusion. (P>0.5).
Line 340: In the “Table 3”, the yield of acetaldehyde when using S. vitulinus 75 is 22.72 ± 2.73. A letter should be added after 22.72 ± 2.73.
Reviewer 6 Report
This is a very good technological and laboratory work showing application of a few staphylococcus strains as starter culture in Turkish dry-cured meat product. The paper is worth publishing after correction:
- definitely English should be checked by native speaker,
-there's a lot of typos (missing spaces) in the text (did anyone read this paper after inserting the text to template?).